# The N-Linked Glycosylation Asn191 and Asn199 Sites Are Controlled Differently Between PKA Signal Transduction and pEKR1/2 Activity in Equine Follicle-Stimulating Hormone Receptor

**DOI:** 10.3390/cimb47030168

**Published:** 2025-03-02

**Authors:** Sung-Hoon Kim, Munkhzaya Byambaragchaa, Sei Hyen Park, Myung-Hum Park, Myung-Hwa Kang, Kwan-Sik Min

**Affiliations:** 1Graduate School of Animal BioScience, Hankyong National University, Anseong 17579, Republic of Korea; bulkkim@gmail.com (S.-H.K.); mrtree119@naver.com (S.H.P.); 2Carbon-Neutral Resources Research Center, Hankyong National University, Aseong 17579, Republic of Korea; munkhzaya_b@yahoo.com; 3Genetic Engineering, Hankyong National University, Anseong 17579, Republic of Korea; 4TNT Research, Sejong 30141, Republic of Korea; pmh@tntresearch.co.kr; 5Department of Food Science and Nutrition, Hoseo University, Asan 31499, Republic of Korea; mhkang@hoseo.edu; 6Division of Animal BioScience, School of Animal Life Convergence Sciences, Hankyong National University, Anseong 17579, Republic of Korea

**Keywords:** equine FSHR, N-glycosylation sites, cAMP response, pERK1/2 activity

## Abstract

Equine follicle-stimulating hormone receptor (eFSHR) contains four extracellular N-linked glycosylation sites, which play important roles in agonist-induced signal transduction. Glycosylation regulates G protein-coupled receptor mechanisms by influencing folding, ligand binding, signaling, trafficking, and internalization. Here, we examined whether the glycosylated sites in eFSHR are necessary for cyclic adenosine monophosphate (cAMP) signal transduction and the phosphate extracellular signal-regulated kinase 1/2 (pERK1/2) response. We constructed mutants (N191Q, N199Q, N268Q, and N293Q) of the four N-linked glycosylation sites in eFSHR using site-directed mutagenesis. In wild-type (wt) eFSHR, the cAMP response gradually increased dose-dependently, displaying a strong response at the EC_50_ and Rmax. Two mutants (N191Q and N199Q) considerably decreased the cAMP response. Both EC_50_ values were approximately 0.46- and 0.44-fold compared to that of the eFSHR-wt, whereas Rmax levels were 0.29- and 0.45-fold compared to eFSHR-wt because of high-ligand treatment. Specifically, the EC_50_ and Rmax values in the N268Q mutant were increased 1.23- and 1.46-fold, respectively, by eFSHR-wt. pERK1/2 activity in eFSHR-wt cells was rapid, peaked within 5 min, consistently sustained until 15 min, and then sharply decreased. pERK1/2 activity in the N191Q mutant showed a pattern similar to that of the wild type, despite impaired cAMP responsiveness. The N199Q mutant showed low pERK1/2 activity at 5 and 15 min. Interestingly, pERK1/2 activity in the N268Q and N298Q mutants was similar to that of eFSHR-wt at 5 min, but neither mutant showed any signaling at 15 min, despite displaying high cAMP responsiveness. Overall, eFSHR N-linked glycosylation sites can signal to pERK1/2 via PKA and the other signals, dependent on G protein coupling and β-arrestin-dependent recruitment. Our results provide strong evidence for a new paradigm in which cAMP signaling is not activated, yet pERK1/2 cascade remains strongly induced.

## 1. Introduction

The follicle-stimulating hormone receptor (FSHR) belongs to the G protein-coupled receptor (GPCR) family, which includes lutropin/choriogonadotropin (LH/CGR) and thyroid-stimulating hormone receptor (THSR), representing the largest family of cell surface proteins involved in signal transduction [1]. These receptors contain a large extracellular domain-binding ligand, seven helical transmembrane domains, and an intracellular domain of the cytoplasmic tail, including several phosphorylation sites [2]. FSH is synthesized and secreted by the pituitary and binds to FSHR, which belongs to class A of the GPCR subfamily. FSHR shows a high degree of ovarian and testis tissue specificity in granulosa and Sertoli cells.

As receptors for a wide range of agonists, GPCRs play a crucial role in many physiological processes [3] and are involved in receptor-mediated signaling pathways, such as the cyclic adenosine monophosphate (cAMP) response [4,5] and phosphate extracellular-related kinase (pERK1/2) activation [6,7]. In FSHRs, many studies have reported that naturally occurring constitutively activating/inactivating mutants cause disease via malfunctioning signal transduction [8,9,10,11].

To better characterize GPCR signaling, many studies have focused on the functions of the post-translational modifications of these receptors [12,13]. Agonist-induced changes in these receptors regulate many essential processes within the cell, such as preparation for folding, trafficking, oligosaccharides, and phosphorylation [14,15]. Even a small error in this process may result in misfolded or dysfunctional receptors.

Specifically, N-linked glycosylation is one of the most common post-translational modifications, involving the sugar modification of asparagine in the consensus amino acid sequence. Many studies have reported that the glycosylation of glycoprotein hormones (FSH, LH, and CG) is important for receptor-mediated signal transduction [16]. In GPCRs, N-linked glycosylation sites are involved in post-translational functions, indicating that the specific glycosylation sites in GPCR significantly influence biological activity, cell surface expression, and internalization.

As indicated by the GPCR results, FSH binding leads to conformational rearrangements within transmembrane regions. These results involve G protein coupling and β-arrestin recruitment. These induce a complex intracellular signaling network [17]. Gαs proteins and β-arrestins are related to extracellular signal-regulated kinase (ERK) signaling via two temporally distinct mechanisms: the rapid-onset G protein-dependent mechanism, and slow-onset β-arrestin-dependent mechanism [18,19]. β-arrestins are involved in phosphate ERK1/2 (pERK1/2) with β-arrestin 1 and 2 involved in FSHR signaling [20,21]. In hFSHR and eel FSHR, pERK1/2 activation occurs within 5 min of FSH stimulation [22,23]. Thus, the specific N-linked glycosylation site of FSHR should be examined to elucidate its involvement in pERK1/2 activation and the cAMP signal response.

In horses, CG secreted by the placenta and LH secreted by the pituitary share identical amino acid sequences and both bind to LH/CGR. However, FSH, which—like LH—is a glycoprotein hormone secreted exclusively by the pituitary, has been scarcely studied in relation to equine FSHR. Recently, as research into the physiological mechanisms of sexual maturation and ovulation in horses has gradually expanded, it has been deemed necessary to investigate the glycosylation function of equine FSHR.

This study aimed to elucidate the signal transduction functions of N-linked glycosylation sites in highly conserved regions of glycoprotein hormone receptors, including FSHR and LH/CGR. We analyzed the cAMP response in the PKA signaling pathway and pEKR1/2 activity.

## 2. Materials and Methods

### 2.1. Materials

Polymerase chain reaction (PCR) kits, including DNA ligation reagents, and restriction enzymes were purchased from Takara Bio (Shiga, Japan). The oligonucleotides used for mutagenesis were synthesized by Genotech (Daejeon, Republic of Korea). The CHO-K1 cells and HEK 293 cells were obtained from the Korean Cell Line Bank (KCLB, Seoul, Republic of Korea). The mammalian expression vector, pCORON1000 SP VSV-G, was purchased from Amersham Biosciences (Piscataway, NJ, USA). Human FSH (hFSH) was purchased from Eurofins Discover X (Fremont, CA, USA). The cAMP Dynamic 2 homogeneous time-resolved fluorescence (HTRF) assay kit was purchased from Cisbio (Codolet, France). The pERK1/2 antibody and total ERK1/2 antibody were sourced from Cell Signaling Technology (Beverly, MA, USA). The QIAprep-Spin plasmid kit was purchased from Qiagen Inc. (Hilden, Germany). All other reagents were purchased from Sigma-Aldrich (St. Louis, MO, USA) and Wako Pure Chemical Industries (Osaka, Japan).

### 2.2. Site-Directed Mutagenesis for N-Linked Glycosylation Sites

To construct point mutations in the four N-linked glycosylation sites, we used template cDNA encoding full-length equine FSHR wild type (eFSHR-wt) using an overlap extension PCR method. The eFSHR cDNA cloned into pcDNA3 and pCORON1000 SP VSV-G was used for mutagenesis, as previously reported [1]. A schematic representation of the four N-linked glycosylation sites (N191, N199, N268, and N293) in the extracellular domain of eFSHR-wt is shown in Figure 1. Each N-linked glycosylation site was changed from Asn (AAC) to Gln (CAG), using an overlapping PCR strategy. Mutagenesis involved a two-step PCR process. First, we amplified fragment I using a combination of the eFSHR forward primer and the targeting mutated reverse primer. Fragment II was amplified using a combination of mutated forward primers and eFSHR reverse primers. The PCR reagents were mixed with 300 ng of DNA template, 2.5 unit of Ex Taq 1 × PCR buffer, 25 mM dNTP, and primers. PCR was performed for 30 cycles using a PCR machine. After the first PCR, fragment I and fragment II products were visualized by agarose gel electrophoresis and purified from the gel using a PCR Clean-up System kit. Subsequently, a second PCR was performed to produce full-length mutants with 300 ng each of fragment I and II, and eFSHR forward and reverse primers. In the second step, the PCR product was purified, the full-length fragments were ligated into the pGEMT-Easy vector, and the plasmids were confirmed by DNA sequencing to confirm the mutated sites. The eFSHR mutants were constructed by introducing the corresponding mutations at positions 191(N191Q), 199 (N199Q), 268 (N268Q), and 293 (n293Q) of the eFSHR cDNA, as shown in Figure 1. Finally, we generated four mutated receptor genes: eFSHR-wt, eFSHR-N191Q, eFSHR-N199Q, eFSHR-N268Q, and eFSHR293Q.

### 2.3. Vector Construction for Transfection in Mammalian Cells

The full-length eFSHR subcloned in the pGEMT-Easy cloning vector was cleaved using the Xho I and Eco RI restriction enzymes, and these sites were introduced into the PCR primers. The fragments were then ligated into the pCORON SP VSVG mammalian expression vector. The ligated eFSHR gene did not contain the eFSHR signal sequence. This was inserted under the VSVG tag into the pCORON SP SVSG vector. The direction of insertion was confirmed by restriction enzyme digestion and sequencing for genetic verification. Finally, five expression vectors were constructed using pVSVG (designated as pVSVG-eFSHRwt, pVSVG-eFSHR-N191Q, pVSVG-eFHSR-N199Q, pVSVG-eFSHR-N268Q, and pVSVG-eFSHR-N293Q).

### 2.4. Transient Transfection into CHO-K1 and HEK 293 Cells

The transient transfection of CHO-K1 and HEK 293 cells was performed using a liposome-mediated transfection method as previously reported [1]. CHO-K1 cells were cultured in growth medium [Ham’s F-12 growth medium supplemented with glutamine (2 mM), 10% fetal bovine serum, and antibiotics]. CHO-K1 cells were passaged in 6-well plates at a density of 1 × 10^6^ cells/mL one day before transfection. HEK 293 cells were cultured in growth medium [Dulbecco’s modified Eagle’s medium containing HEPES (10 mM), gentamycin (50 μg/mL), and 10% fetal bovine serum]. For transfection, HEK 293 cells were seeded in 6-well plates, grown to 80–90% confluence on the day of transfection.

Each plasmid DNA (2.5 µg) and transfection reagent (5 μL) was diluted with Opti-MEM-reduced Serum Medium to a final volume of 250 μL. After incubation for 5 min at room temperature, the diluted plasmid DNA and reagent were mixed and incubated for 20 min at room temperature. Complexes were added to the cells dropwise after washing twice with phosphate-buffered saline or Opti-MEM buffer. The cells were then incubated at 37 °C in a 5% CO_2_ incubator. After 5 h, fresh growth medium containing 20% fetal bovine serum was added to each well. CHO-K1 cells were used to measure the cAMP response at 48–72 h post-transfection. HEK 293 cells were subjected to a pERK1/2 activity assay.

### 2.5. Analysis of cAMP Levels via Homogeneous Time-Resolved Fluorescence Assays

cAMP accumulation in CHO-K1 cells expressing eFSHR-wt or its mutants was measured using cAMP Dynamic 2 competitive immunoassay kits, as previously reported [1]. Briefly, the transfected cells were diluted in 0.5 mM 3-isobutyl-1-methylxanthine (IBMX) to inhibit cAMP degradation and seeded in 383-well plates at a density of 1 × 10^4^ cells/well. The cells were then stimulated with hFSH (5 μL) in a dose-dependent manner for 30 min at room temperature. Standard samples were prepared to cover an average cAMP concentration (0.17–712 nM). The d2-labeled cAMP reagent (5 μL: diluted 5 times in lysis buffer) and anti-cAMP cryptate-conjugated monoclonal antibody (5 μL: diluted 5 times in lysis buffer) were added to each well, followed by incubation for 1 h at room temperature.

cAMP was detected by measuring the decrease in homogeneous time-resolved fluorescence (HTRF) energy transfer (665 nm/620 nm) using a TriStar LB 942 microplate reader (BERTHOLD Tech, Wildbad, Germany). This method is a competitive immunoassay between native cAMP produced by cells and cAMP labeled with dye d2. Tracer binding was visualized using an anti-cAMP monoclonal antibody labeled with Eu^3+^-Cryptate. The specific signal Delta F% (energy transfer) was inversely proportional to the cAMP concentration in the standard or sample.

The results were calculated from the ratio of fluorescence at 665 and 620 nm, expressed as Delta F% (cAMP inhibition), according to the following equation: Delta F% = [(standard or sample ratio-negative ratio) × 100]/negative ratio. The cAMP concentration for Delta F% values was calculated using GraphPad Prism software (version 6.0; GraphPad Software Inc., La Jolla, CA, USA).

### 2.6. Phospho-ERK1/2 Time Course

For the ERK phosphorylation assay, HEK 293 cells were transfected with pVSVG plasmids containing the wt and mutant eFSHR sequences. After 48 h, the cells were starved for at least 6 h and stimulated with hFSH (500 ng/mL) in a time-dependent manner. After washing the cells with ice-cold PBS, the ice-cold lysis buffer was added. The cells were then scraped off using cold plastic scrapers and collected in microcentrifuge tubes. The cells were agitated with ice-cold lysis buffer for 30 min at 4 °C. The tubes were then centrifuged at 16,000× *g* for 20 min at 4 °C. The supernatants were then collected in fresh tubes and placed on ice. The protein concentration in the cell lysates was determined using the Bradford assay (Bio-Rad Laboratories, Richmond, CA, USA).

Equal amounts (20 μg) of protein samples were loaded onto 12% SDS-PAGE gels and transferred onto polyvinylidene difluoride (PVDF) nitrocellulose membranes using a Bio-Rad Mini Trans-Blot electrophoresis cell (Hercules, CA, USA). The membrane was blocked with 5% skim milk solution in TBS-T (20 mM Tris-HCl, pH 7.6, 140 mM NaCl, 0.1% Tween 20) for 1 h on a shaker at RT. pERK1/2 and total ERK1/2 were detected via immunoblotting using rabbit polyclonal anti-phospho-p44/42 MAPK (1:2000) and anti-MAPK1/2 (1:3000) overnight at 4 °C. Membranes were then incubated with horseradish peroxidase-labeled anti-rabbit secondary antibodies for 1 h. The membranes were washed 3 times with TBS-T. Chemiluminescence was performed using SuperSignal West Femto maximum sensitivity substrate, and pERK1/2 immunoblots were quantified via densitometry using Image Lab v.6.0 (Bio-Rad, Hercules, CA, USA).

### 2.7. Data Analysis

Sequence alignment was performed using the MultAlin multiple sequence analysis tool. Dose–response curves for the cAMP response were generated for experiments performed in duplicate. GraphPad Prism software (version 6.0; San Diego, CA, USA) was used to analyze the cAMP response, EC_50_ values, and stimulation curves. Curves fitted in a single experiment were normalized to background signals measured in mock-transfected cells. The pERK1/2 values measured by densitometry were drawn using GraFit (version 5.0; Erithacus Software, Horley, Surrey, UK). Data are presented as means ± SEM of three or four biological replicates. Statistical significance was determined by one-way analysis of variance (ANOVA), followed by Tukey’s comparison test using GraphPad Prism v.6.0, as indicated in the figure captions. Statistical significance was determined at the following levels: * *p* < 0.05, compared to eFSHR-wt cells at the corresponding time point.

## 3. Results

### 3.1. Construction of eFSHR Glycosylation Mutants

To determine the effect of the N-linked glycosylation sites of eFSHR on ligand–receptor interactions, four mutants (N191Q, N199Q, N268Q, and N298Q) were constructed in the extracellular domain of eFSHR, as shown in Figure 1. These sites are conserved in most mammalian FSHR, including donkey, human, rat, and mouse FSHR. As described in the Materials and Methods section, we constructed four mutants and cloned them into the pCORON1000 SP-VSVG mammalian expression vector without the eFSHR signal sequence. It is known that residue 191 is located in exon 6, while residue 199 is located in exon 7 of FSHR. We designated the plasmids pVSVG-eFSHR-N191Q, N199Q, N268Q, and N298Q.

### 3.2. cAMP Response for eFSHR N-Glycosylation Mutants

In vitro biological activity was assessed using transfected CHO-K1 cells expressing eFSHR. The effects of the glycosylated mutants on basal and ligand-stimulated cAMP responsiveness are summarized in Figure 2 and Table 1. The potency of cAMP activation in eFSHR-wt cells increased in a dose-dependent manner upon agonist treatment (Figure 2).

The Rmax cAMP level in the wild-type receptor was 108.5 nM/10^4^ cells and the half-maximal effective concentration (EC_50_) of ligand-stimulated cAMP was approximately 989.5 ng/mL, indicating complete biological activity in this system. In cells expressing the N191Q and N199Q mutants, the plots shifted to the right compared to those of eFSHR-wt, displaying exceptionally low EC_50_ values of approximately 2106 ng/mL and 2206 ng/mL, respectively. These values were approximately 0.46- and 0.44-fold those of eFSHR-wt. The Rmax values in both mutants were 31.1 nM and 48.7 nM, approximately 29% and 45% compared to those of eFSHR-wt, respectively (Table 1 and Figure 3).

In particular, the EC_50_ value in cells expressing the N268Q mutant was 805 ng/mL, indicating a 1.23-fold increase compared to the wild type. The Rmax levels in this mutant also increased to 159.2 nM/10^4^ cells, representing a 1.46-fold increase compared to eFSHR-wt. The EC_50_ value of the other mutant (N293Q) decreased slightly to 1243 ng/mL, which was 0.79-fold that of the eFSHR-wt. However, the Rmax level was only 0.93-fold compared with that of the eFSHR-wt. In the present study, specific N-glycosylation sites were found to play a particularly key role in the PKA signal transduction pathway, indicating that the cAMP response was only less than 50% in the N191Q and N199Q mutants. Thus, we suggest that these glycosylation sites are involved in signal transduction, as indicated by decreased EC_50_ and Rmax levels. Therefore, N191 and N199 glycosylation sites of eFSHR play a critical role in receptor-mediated cAMP responsiveness. In a previous study, we suggested that N-linked glycosylation sites (N120Q, N191Q, N272Q, and N288Q) in eel FSHR are important in PKA signal transduction. The same mutant (N191Q) in eel FSHR completely impaired the cAMP response, indicating involvement in in cell surface expression [2].

### 3.3. Phospho-ERK1/2 Activation Stimulated by eFSHR in HEK293 Cells

HEK 293 cells transiently transfected to express eFSHR-wt and stimulated with FSH showed a time-dependent increase in MAP kinase activation (Figure 4). An early and rapid increase in ERK1/2 levels was observed at 5 min. pERK1/2 activation peaked at 5 min and was consistently maintained until 15 min; however, its activity abruptly decreased to basal levels at 30 min (Figure 4A,D). In the N191Q mutant, the pERK1/2 activity was similar to that of eFSHR-wt at 5 min. However, it decreased slightly by 35% of that at 15 min and then completely decreased to basal levels at 30 min (Figure 4A,D).

Interestingly, pERK1/2 activity in cells expressing the N199Q mutant was approximately 34.8% at 5 min compared to that in the wild type, but none of the other mutants showed any decrease at 5 min. At 15 min, pERK1/2 increased slightly and was completely reduced to the basal levels at 30 min, similar to that observed in the eFSHR-wt and N191Q mutants (Figure 4B,E). The N191Q and N199Q mutants showed a decrease to 63% and 40% in pERK1/2 levels at 15 min, respectively. Therefore, the N191Q mutation seems to play an important role in mediating the activation of pERK1/2 signaling, which is induced by FSH, as observed around the 15 min time point. The N199Q mutant showed low pERK1/2 activity at 5–15 min, but signaling still occurred.

The other two mutants (N268Q and N293Q) showed the same pERK1/2 activity at 5 min, but this decreased to 7% and 0% with a rare signal at 15 min compared to that of the wild type, as shown in Figure 4C,F,G. None of the mutant receptors, including eFSHR-wt, showed pERK1/2 signaling at 30 min. Although two mutants (N191Q and N199Q) displayed low EC_50_ value and Rmax level for the cAMP response, pERK1/2 activity was confirmed. Thus, another signaling pathway, excluding cAMP signaling, may be used by the two mutants (N191Q and N199Q) [9,10]. GRKs and β-arrestin pathways may be potential substitutes for cAMP signaling.

## 4. Discussion

Studies on GPCRs have shown that their post-translational modifications affect indispensable regulatory functions, such as folding, trafficking, and signaling. In this study, we described the signal transduction of N-linked glycosylation sites in the extracellular domain of equine FSHR. We found that four glycosylation sites (N191, N199, N296, and N293) were located in a highly conserved region among glycoprotein hormone receptors. Our findings demonstrated that the N191Q at the specific glycosylation site dramatically impaired cAMP signaling, but not pERK1/2 activation. In the N199Q mutation, both signaling pathways were decreased compared with those in eFSHR-wt. In the other two mutants (N296Q and N293Q), the cAMP response was increased or similar to that of eFSHR-wt, and pERK activity was also similar at 5 min, but no signaling appeared at 15 min after agonist treatment. Therefore, the effect of PKA signaling on pERK at eFSHR N-linked glycosylation sites is assumed to differ. These results suggest that the highly conserved N-linked glycosylation regions in glycoprotein hormone receptors independently influence the downstream signal transduction for pERK1/2 activation.

In this study, we characterized signal transduction at four N-linked glycosylation sites in the extracellular domain, which is highly conserved in mammalian eFSHR. PKA signaling dramatically attenuated the agonist-induced cAMP responsiveness in cells expressing the N191Q mutant. The EC_50_ and Rmax values were only 0.46- and 0.29-fold, respectively, compared to those of eFSHR-wt, demonstrating that the N191 glycosylation site impairs PKA signal transduction. These results are consistent with those of a previous study showing that the N174Q mutation in rat FSHR completely impaired its function and that this site was required for receptor folding to provide high-affinity binding [24,25]. We also reported that the same mutation (N191Q) in eel FSHR resulted in completely impaired signal transduction. In rat LH/CGR, the same conserved region (N173Q) is important for PKA signaling, indicating that this site is involved in the high-affinity binding of LH [26].

We found that the mutation of the N199 residue significantly decreased the cAMP response, indicating that the EC_50_ and Rmax values were 0.44- and 0.45-fold, respectively, compared with those of eFSHR-wt. However, their responsiveness was not completely impaired. Further, the mutation of N268 resulted in an increased functional response to cAMP. In addition, the N293Q mutant exhibited a slight decrease in cAMP signaling. These two glycosylation sites are located close to the transmembrane domain. Therefore, this glycosylation residue may impede the flexibility of this domain, thereby inhibiting signal transmission to the membrane domain.

Many studies on FSHR mutations, including SNPs, lack full documentation on receptor bias. However, the Ala189Val mutation impairs G protein signaling and reduces plasma membrane expression by promoting intracellular retention [9,10,17]. The inactive Ala189Val mutation is most closely related to the N191 glycosylation site. Conformational changes around these sites did not trigger the cAMP signal transduction pathway. Based on the results of the present study, the N199Q mutation impairs the PKA signal transduction pathway. Therefore, we suggest that, among the N-linked glycosylation sites of eFSHR, N191may significantly influence cAMP signaling.

The intracellular signaling pathway is activated by FSH agonists and includes the Gs-mediated activation of adenylate cyclase, showing an increase in cAMP responsiveness and PKA activation in a G protein-dependent manner [27]. pERK1/2 proceeds via the sequential activation of three kinases: Raf1, MEK1, and ERK1/2 [28,29]. Similar to most GPCRs, the FSHR interacts with β-arrestins, known for their key role in receptor desensitization and internalization, and it can activate signaling pathways in response to FSH agonist stimulation [20,30]. This ERK1/2-mediated regulatory process has been observed for many GPCRs interacting with β-arrestins, including β2 adrenergic receptor, u-opioid receptor [31], and glucagon-like peptide-1 receptor [32].

In the present study, pERK1/2 activity exhibited a peak response 5 min after FSH agonist stimulation of all receptors. These results are consistent with FSH-, LH- and hCG-stimulated pERK1/2 activation in a time-dependent manner, with the peak observed at approximately 6 min [23,27]. In the mutation data, the pERK1/2 activity for the N191Q mutation was similar to that of eFSHR-wt within 5 min, despite having the lowest cAMP responsiveness. However, the N199Q mutation significantly decreased at 5 and 15 min after FSH agonist treatment. The N199Q mutation dramatically decreased PKA signaling. Therefore, this conclusion can be assumed. However, the N191Q mutation did not explain the relationship between cAMP responsiveness and pERK1/2 activity. The activity of the other two mutations (N268Q and N293Q) peaked after 5 min, but completely decreased to 0% in 15 min. The cAMP responsiveness in these two mutants increased or decreased slightly compared to that of eFSHR-wt. In particular, the Rmax value increased significantly.

Our results are consistent with those of previous studies, wherein hFSHR-M512I mutation with spontaneous ovarian syndrome showed a significant reduction in both cAMP and phosphatidylinositol-3 kinase phosphorylation with unchanged pERK1/2 activity [10], and a partially deglycosylated equine LH could selectively activate β-arrestin-dependent signaling pathways despite lacking cAMP responsiveness at the FSHR [33]. Many studies have shown that G protein- and β-arrestin-dependent signaling pathways are independent. Specific GPCRs selectively trigger β-arrestin-dependent signaling pathways without the occurrence of G protein coupling [34]. Some GPCR ligands selectively activate β-arrestin-mediated signaling [35,36]. In binding assays, hypoglycosylated FSH21/18, missing either Asn7 or Asn24 N-linked oligosaccharide on the β-subunit, was found to be 9–26-fold more active than fully glycosylated FSH24 [37], but hyperglycosylated FSH increased the ovulated eggs and in vitro embryo development [38]. Thus, FSH variants are involved in the biased signaling between FSH21/18 and FSH24. Based on previous research findings and the results of this study, it is speculated that cAMP and β-arrestin signaling in FSHR are regulated independently.

In recent studies on glycoprotein hormone receptors, cryo-electron microscopy structures of the FSH–FSHR and CG–LH/CGR complexes have provided insights into hormone specificity and hormone-induced receptor activation [39,40]. A conserved activation mechanism has been identified, in which the unique residue H615 in FSHR plays a key role in determining the receptor’s response to various allosteric agonists. Moreover, the allosteric agonist Org43553 acts on LH/CGR by binding to a highly conserved 10-residue fragment located between the extracellular and transmembrane domains. Therefore, it is believed that in these two mutants (N191Q and N199Q), conformational changes resulting from post-translational modifications significantly affect receptor–ligand interactions and downstream signaling.

Based on our results, the N-linked glycosylation sites in eFSHR are independent of cAMP signaling and pERK1/2 activity. The N191 glycosylation site plays an indispensable role in PKA signaling but not in pERK1/2 activity. The specific N-linked glycosylation site (N199) plays a significant role in both signaling pathways. The other two N-linked glycosylation sites (N268 and N293) are not cAMP-responsive, but are necessary for pERK1/2 activity upon FSH agonist treatment. Taken together, these results indicate that the stimulation of the eFSHR-mediated signaling pathway independently leads to pERK1/2 activation. In specifically biased FSHR signaling, receptor-mediated pERK1/2 pathways must be studied further to determine the presence of a new signaling pathway involving the phosphorylation of downstream effectors of the MAPK pathway.

## 5. Conclusions

Our study clearly demonstrated that mutations in the N-linked glycosylation sites of eFSHR independently regulate intercellular signaling between the PKA signal and pERK1/2 activity. The N191 glycosylation site is crucial for maintaining cAMP responsiveness, whereas pERK1/2 activity remains unaffected by its absence, indicating that this glycosylation site plays a specific and indispensable role in modulating the cAMP signaling pathway. The N199 glycosylation site is essential for the proper functioning of both cAMP and pERK1/2 signaling pathways, suggesting its critical role in maintaining receptor-mediated signal transduction and ensuring balanced intracellular responses. Thus, the N-linked glycosylation sites in eFSHR are selectively regulated by the PKA signaling pathway and pERK1/2 activity. Therefore, these two mutants could serve as valuable models for conducting a systematic study on biased signaling, including both the cAMP pathway and pERK1/2 signaling, providing insights into the mechanisms underlying receptor-specific signal regulation.

## Figures and Tables

**Figure 1 cimb-47-00168-f001:**
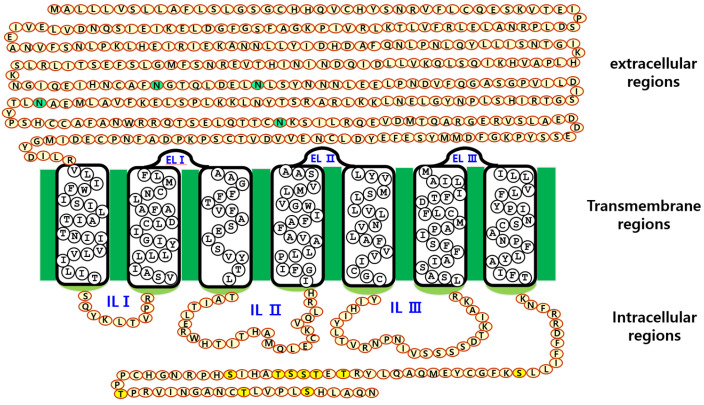
Schematic representation of the eFSHR structure. The N-linked glycosylation sites (N191, N199, N268, and N293) in the extracellular domain regions are indicated. The green circles indicate the putative four N-glycosylated sites. The extracellular region comprises 365 amino acids, indicating the longest extracellular region among G protein-coupled receptors. The three intracellular loops comprise 9, 25, and 25 amino acids. The intracellular regions have 65 amino acids, and the 10 potential phosphorylation sites (serine and threonine residues) are S641, T655, T657, S658, S659, T660, S664, T674, T684, and S689, indicated by yellow circles. EL, extracellular loop; IL, intracellular loop.

**Figure 2 cimb-47-00168-f002:**
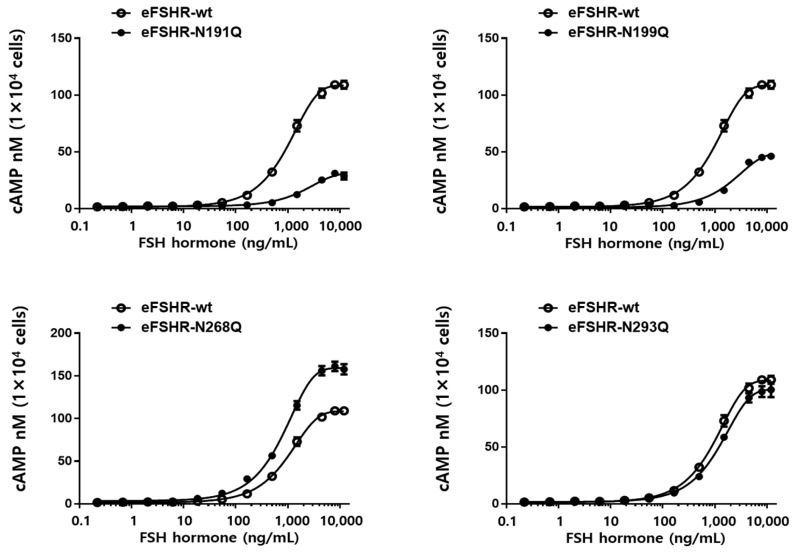
Total cAMP levels induced by stimulation with the FSH in CHO-K1 cells transiently transfected with the N-linked glycosylation site mutants of eFSHR. Empty circles denote eFSHR-wt and black circles denote each mutant. The value of ΔF% was recalculated as cAMP concentration (nM). A representative dataset was obtained from three independent experiments. The figure depicts the results of a representative experiment performed with the indicated mutants.

**Figure 3 cimb-47-00168-f003:**
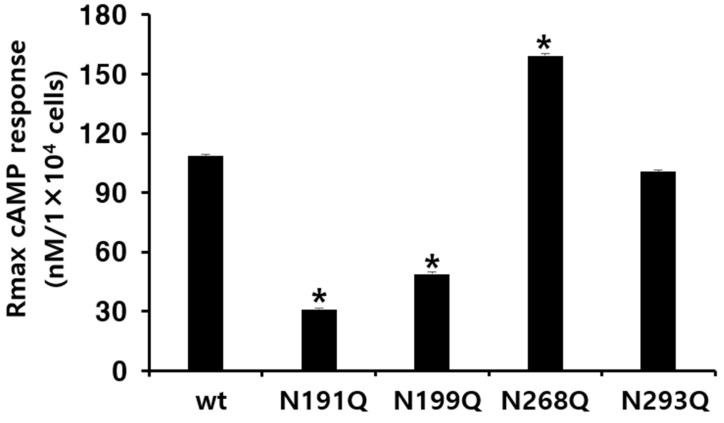
Rmax levels in the N-linked glycosylation mutants of eFSHR. The maximal cAMP responses presented in Figure 2 are displayed using a bar graph. * Statistically significant differences (*p* < 0.05) compared to the Rmax level of the eFSHR wild type.

**Figure 4 cimb-47-00168-f004:**
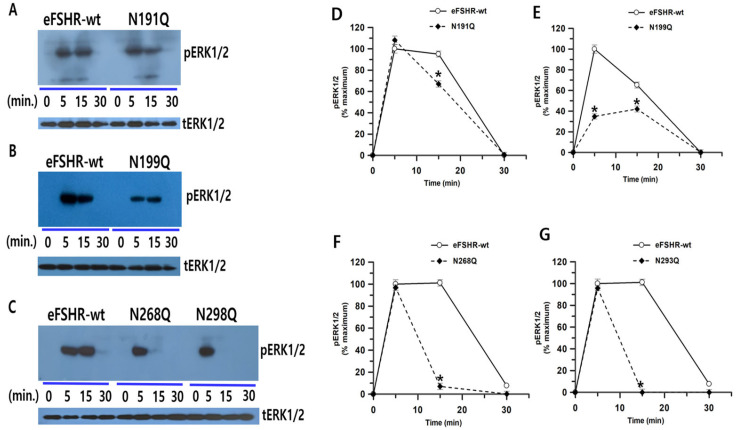
pERK1/2 activation in HEK-293 cells transfected with eFSHR-wt and mutants after stimulation with FSH. HEK-293 cells transiently expressing eFSHR-wt or mutants were serum-starved for 4–6 h before stimulation with 500 ng/mL FSH for the indicated time periods. Whole-cell lysates were analyzed for pERK1/2 and total ERK levels. (**A**–**C**) Western blot results for phospho-ERK1/2 and total ERK. (**D**–**G**) Quantified phosphor-ERK1/2 levels, normalized to total ERK, are expressed as a percentage of the maximal response (100% for eFSHR-wt at 5 min). Densitometry was used to quantify the phospho-ERK1/2 band. Representative data are shown, and graphs represent the mean ± SEM of three independent experiments. Statistical significance was determined using one-way ANOVA, followed by Tukey’s comparison test. * *p* < 0.05 compared with eFSHR-wt cells at the corresponding time point.

**Table 1 cimb-47-00168-t001:** Bioactivity of equine FSHR mutants.

	cAMP Responses		
eFSH receptors	Basal *^a^* (nM/10^4^ cells)	Log (EC_50_) (ng/mL)	Rmax *^b^* (nM/10^4^ cells)
eFSHR-wt	1.5 ± 0.4	989.5 (1.0-fold) (929 to 1058) *^c^*	108.5 ± 0.8 (1-fold)
eFSHR-N191Q	1.9 ± 0.3	2106 (0.46-fold) (1917 to 2598)	31.1 ± 0.3 (0.29-fold)
eFSHR-N199Q	0.9 ± 0.4	2206 (0.44-fold) (2124 to 2367)	48.7 ± 1.1 (0.45-fold)
eFSHR-N268Q	3.4 ± 0.7	805.9 (1.23-fold) (757 to 861)	159.2 ± 1.2 (1.46-fold)
eFSHR-293Q	1.7 ± 0.5	1243 (0.79-fold) (1157 to 1344)	100.5 ± 0.9 (0.93-fold)

Values are the means ± SEM of triplicate experiments. Log (EC_50_) values were determined from the concentration–response curves obtained from in vitro bioassays. *^a^* Basal cAMP level average without agonist treatment. *^b^* Rmax average cAMP level/10^4^ cells. *^c^* Geometric mean (95% confidence limit) of at least three experiments.

## Data Availability

All relevant data are contained within the article.

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
