# Peer review of "The N-Linked Glycosylation Asn191 and Asn199 Sites Are Controlled Differently Between PKA Signal Transduction and pEKR1/2 Activity in Equine Follicle-Stimulating Hormone Receptor"

_cimb, 2025, doi:10.3390/cimb47030168_

Round 1
Reviewer 1 Report
Comments and Suggestions for Authors
This is another similar study by the authors, where they have analyzed the four extracellular N-linked glycosylation sites in the equine follicle-stimulating hormone receptor compared to previously reported in eel-FSHR. They have examined whether these sites are necessary for cAMP signal transduction and pERK1/2 response. They have concluded that the N191 glycosylation site is indispensable for cAMP response, while N199 is necessary for both cAMP response and pERK1/2 activity.
I have several concerns related to this study.
- The authors have made mutant clones but did not confirm the expression levels of these mutants relative to the wt clone. There are possibilities that mutation at certain places in the protein sequence results in protein instability and thus a lower level of expression, or a short-term expression and sometimes truncated expression. These clones have been used to show the pERK1/2 activity, highlighting the activity of these mutant surface receptors compared to the wt. Therefore, ascertaining the expression of these mutants is necessary. In addition, if there is a loss of surface receptors, they should follow the time course for which they have analyzed the pERK1/2 activation with respect to the protein expression of the mutants.
- This study could have been better if the authors had made double or triple mutants in addition to single mutants for the e-FSHR and analyzed for the signaling. This approach will specifically highlight the importance of each site separately in the presence or absence of other sites.
- It is good to present elaborate material and methods, but repeating the same in the results is not advisable. In the result section 3.1, the authors have almost repeated the methodology.
- Authors are suggested to modify the result section 3.3 according to the figure referred to. There are misinterpretations of western blot quantification data at all places in the paragraph. For e.g., “However, it decreased slightly to 35% of that at 15 min and then completely decreased to basal levels at 30 min.” It should be modified to “slightly by 35%.". “The other two mutants (N268Q and N293Q) showed the same pERK1/2 activity at 5 min, but this decreased by approximately 7% and 0% with a rare signal at 15 min compared to that of the wild type." It should be decreased to 7%.
- Line 318: “Thus, N191Q was appears to mediate FSH-mediated pERK1/2 signaling of eFSHR at 15 min.” What does the author mean by this statement?
- Line 326: “Thus, another signaling pathway, excluding cAMP signaling, may be used by the two mutants (N191Q and N199Q). GRKs and b-arrestin pathways may be potential substitutes for cAMP signaling.” Authors should cite some references substantiating the possibility of other pathways.
- In table 1, the log EC50 value for eFSHR-N199Q is 2206, while its geometric mean value ranges between 12.4 and 16.7. How is this possible?
Comments on the Quality of English Language
Considerable english improvement is required in the result section.
Author Response
This is another similar study by the authors, where they have analyzed the four extracellular N-linked glycosylation sites in the equine follicle-stimulating hormone receptor compared to previously reported in eel-FSHR. They have examined whether these sites are necessary for cAMP signal transduction and pERK1/2 response. They have concluded that the N191 glycosylation site is indispensable for cAMP response, while N199 is necessary for both cAMP response and pERK1/2 activity.
I have several concerns related to this study.
- The authors have made mutant clones but did not confirm the expression levels of these mutants relative to the wt clone. There are possibilities that mutation at certain places in the protein sequence results in protein instability and thus a lower level of expression, or a short-term expression and sometimes truncated expression. These clones have been used to show the pERK1/2 activity, highlighting the activity of these mutant surface receptors compared to the wt. Therefore, ascertaining the expression of these mutants is necessary. In addition, if there is a loss of surface receptors, they should follow the time course for which they have analyzed the pERK1/2 activation with respect to the protein expression of the mutants.
- →Reviewer 2 raised the same comment. It is true that we did not present analyses of cell surface expression levels or cell surface loss experiments. However, Ref. #2 provided clear data on this aspect for eel FSHR. Therefore, we did not conduct these expression experiments in our study.
- This study could have been better if the authors had made double or triple mutants in addition to single mutants for the e-FSHR and analyzed for the signaling. This approach will specifically highlight the importance of each site separately in the presence or absence of other sites.
- →At the current stage, we have presented only the functional analyses for each glycosylation site, and we are considering conducting experiments on the double mutant at positions 191 and 199 in the future.
- It is good to present elaborate material and methods, but repeating the same in the results is not advisable. In the result section 3.1, the authors have almost repeated the methodology.
- →We deleted “repeating part” in the 3.1 Section.
- Authors are suggested to modify the result section 3.3 according to the figure referred to. There are misinterpretations of western blot quantification data at all places in the paragraph. For e.g., “However, it decreased slightly to 35% of that at 15 min and then completely decreased to basal levels at 30 min.” It should be modified to “slightly by 35%.". “The other two mutants (N268Q and N293Q) showed the same pERK1/2 activity at 5 min, but this decreased by approximately 7% and 0% with a rare signal at 15 min compared to that of the wild type." It should be decreased to 7%.
- →We changed by reviewer’s comment.
- Line 318: “Thus, N191Q was appears to mediate FSH-mediated pERK1/2 signaling of eFSHR at 15 min.” What does the author mean by this statement?
- →Therefore, the N191Q mutation seems to play an important role in mediating the activation of pERK1/2 signaling, which is induced by FSH, as observed around the 15-minute time point.
- Line 326: “Thus, another signaling pathway, excluding cAMP signaling, may be used by the two mutants (N191Q and N199Q). GRKs and b-arrestin pathways may be potential substitutes for cAMP signaling.” Authors should cite some references substantiating the possibility of other pathways.
- →we inserted “Ref #9,10.
- In table 1, the log EC50 value for eFSHR-N199Q is 2206, while its geometric mean value ranges between 12.4 and 16.7. How is this possible?
- →We change “12.4 and 16.7” to “2124 to 2367”.

Reviewer 2 Report
Comments and Suggestions for Authors
The authors present a study investigating the role of four (potential) extracellular N-linked glycosylation sites in the equine follicle-stimulating hormone receptor (eFSHR) on signal transduction pathways, specifically cAMP production and pERK1/2 activation. While the study provides data on the functional impacts of these N-glycosylation sites, similar findings have already reported. The manuscript lacks intrinsic novelty, and the authors need to focus more on comparative analysis by integrating amino acid sequence and 3D structural insights.
1) Several similar studies papers have been published, including their own studies (e.g. ref #2, #22). I suggest summarizing the current findings alongside previous results in a comparative table and discussing them in detail based on amino acid sequence and 3D structure. For example, the N268Q mutant of equine FSHR exhibited a higher Rmax than the equine WT, whereas the N272Q mutant of eel FSHR showed a lower Rmax than eel WT (ref #2). Given that the 3D structure of FSHR is now available (Dual et al. Nat Commun 14, 2023), the authors should leverage structural insights to explain these differences. Without such analysis, the manuscript offers only minor variations from previously published studies.
2) The rationale for selecting the “equine” receptor remains unclear. Why was this particular species chosen? The authors should provide an explanation for its relevance.
3) No data are provided on the expression level of wild-type and mutant receptors. Since variations in receptor expression could influence the observed signaling responses, it is essential to clarify whether expression levels were measured and how the results were normalized.
4) The manuscript assumes that all four potential N-glycosylation sites are occupied by N-glycans, but this is not experimentally validated here. Have the authors confirmed N-glycosylation at these sites using techniques such as PNGase F treatment, mass spectrometry or lectin blotting? Such validation is crucial to support the functional conclusions drawn from mutagenesis experiments.
5) The study lacks a detailed mechanistic explanation of how glycosylation influences eFSHR signaling. The authors are encouraged to incorporate insights from the reported 3D structure of FSHR to discuss potential confirmation changes, receptor-ligand interactions, and downstream signaling effects.
Comments on the Quality of English Language
English is good.
Author Response
Reviewer 2
Open Review
Comments and Suggestions for Authors
The authors present a study investigating the role of four (potential) extracellular N-linked glycosylation sites in the equine follicle-stimulating hormone receptor (eFSHR) on signal transduction pathways, specifically cAMP production and pERK1/2 activation. While the study provides data on the functional impacts of these N-glycosylation sites, similar findings have already been reported. The manuscript lacks intrinsic novelty, and the authors need to focus more on comparative analysis by integrating amino acid sequence and 3D structural insights.
- Several similar studies papers have been published, including their own studies (e.g. ref #2, #22). I suggest summarizing the current findings alongside previous results in a comparative table and discussing them in detail based on amino acid sequence and 3D structure. For example, the N268Q mutant of equine FSHR exhibited a higher Rmax than the equine WT, whereas the N272Q mutant of eel FSHR showed a lower Rmax than eel WT (ref #2). Given that the 3D structure of FSHR is now available (Dual et al. Nat Commun 14, 2023), the authors should leverage structural insights to explain these differences. Without such analysis, the manuscript offers only minor variations from previously published studies.
→ Of course, there have been reports on the functional analysis of N-linked glycosylation in rat FSHR (Ref #24), and in our laboratory we also reported the functional analysis of glycosylation in eel FSHR (Ref #2). However, for rat FSHR, an analysis of pERK1/2 has not been conducted, and in the eel study, the pERK1/2 activity was not reported. Therefore, as indicated in the title, it was confirmed that N-linked glycosylation (N195 located at exon 7) specifically functions in cAMP regulation in both rat and eel, while the pERK1/2 signaling, although slightly lower, proceeds normally, this being a major point of the study. In addition, the conclusions have been supplemented with the discussion of structural analyses of FSHR and LH/CGR from Duan et al. (2021, 2023), presented the activated and inactivated 3D structures produced using an insect cell expression system.
- The rationale for selecting the “equine” receptor remains unclear. Why was this particular species chosen? The authors should provide an explanation for its relevance.
→We inserted in the Introduction section (Line 103-108) “In horses, CG secreted by the placenta and LH secreted by the pituitary share identical amino acid sequences and both bind to LH/CGR. However, FSH, which—like LH—is a glycoprotein hormone secreted exclusively by the pituitary, has been scarcely studied in relation to equine FSHR. Recently, as research into the physiological mechanisms of sexual maturation and ovulation in horses has gradually expanded, it has been deemed necessary to investigate the glycosylation function of equine FSHR.”
- No data are provided on the expression level of wild-type and mutant receptors. Since variations in receptor expression could influence the observed signaling responses, it is essential to clarify whether expression levels were measured and how the results were normalized.
→In accordance with the reviewer’s comment, expression-related analyses were not performed in this study. In reference #2, cell surface expression levels were examined, and the variant present in exon 7 exhibited significantly lower expression. Additionally, although we are currently preparing another manuscript, our research on equine LH/CGR and eel LH/CGR has already demonstrated that the glycosylation site in exon 7 consistently has a significantly negative effect on both cAMP activity and expression levels. Therefore, due to time constraints, expression experiments for equine FSHR were not conducted, and the manuscript was submitted in its present form.
- The manuscript assumes that all four potential N-glycosylation sites are occupied by N-glycans, but this is not experimentally validated here. Have the authors confirmed N-glycosylation at these sites using techniques such as PNGase F treatment, mass spectrometry or lectin blotting? Such validation is crucial to support the functional conclusions drawn from mutagenesis experiments.
→As the reviewer noted, it is very important to analyze which glycosylation sites undergo proper post-translational modification. However, in our previous studies, we performed Western blot experiments on several receptors, but the cell surface expression levels were too low to yield results of sufficient quality for presentation. Therefore, no experimental results for this aspect are included.
- The study lacks a detailed mechanistic explanation of how glycosylation influences eFSHR signaling. The authors are encouraged to incorporate insights from the reported 3D structure of FSHR to discuss potential confirmation changes, receptor-ligand interactions, and downstream signaling effects.
→ We inserted in the discussion section (Line 417-426) In recent studies on glycoprotein hormone receptors, cryo-electron microscopy structures of the FSH–FSHR and CG–LH/CGR complexes have provided insights into hormone specificity and hormone-induced receptor activation (Duan et al., 000). A conserved activation mechanism has been identified, in which the unique residue H615 in FSHR plays a key role in determining the receptor’s response to various allosteric agonists. Moreover, the allosteric agonist Org43553 acts on LH/CGR by binding to a highly conserved 10-residue fragment located between the extracellular and transmembrane domains. Therefore, it is believed that in these two mutants (N191Q and N199Q), conformational changes resulting from post-translational modifications significantly affect receptor-ligand interactions and downstream signaling.

Reviewer 3 Report
Comments and Suggestions for Authors
Thank you for opportunity to review this manuscript.
The follicle-stimulating hormone receptor (FSHR) is a member of the G protein-coupled receptor (GPCR) superfamily, a vast and diverse group of cell surface proteins critical for signal transduction. GPCRs, including FSHR, are vital for mediating cellular responses to a wide array of agonists, thereby playing a central role in numerous physiological processes. They initiate receptor-mediated signaling cascades, notably the cyclic adenosine monophosphate (cAMP) pathway and the activation of phosphate extracellular signal-regulated kinases 1 and 2 (pERK1/2). Disruptions in these signaling pathways, often stemming from naturally occurring constitutively activating or inactivating mutations in FSHR, have been implicated in various diseases. In essence, the proper functioning of FSHR is essential for maintaining cellular homeostasis, and aberrations in its structure or signaling capabilities can have significant pathological consequences. Using site-directed mutagenesis, researchers created eFSHR mutants with mutations at each of the four N-linked glycosylation sites (N191Q, N199Q, N268Q, and N293Q). Wild-type (wt) eFSHR exhibited a dose-dependent increase in cAMP response, with a strong response observed at the EC50 and Rmax. In contrast, the N191Q and N199Q mutants showed a significant reduction in cAMP response. Compared to eFSHR-wt, the EC50 values for these mutants were approximately 0.46- and 0.44-fold, respectively, while the Rmax levels were 0.29- and 0.45-fold lower, respectively, even with high-ligand treatment. Conversely, the N268Q mutant displayed increased cAMP responsiveness, with EC50 and Rmax values increased by 1.23- and 1.46-fold, respectively, compared to eFSHR-wt. In eFSHR-wt cells, pERK1/2 activity was rapid, peaking within 5 minutes and remaining sustained until 15 minutes before declining sharply. The N191Q mutant exhibited a similar pERK1/2 activation pattern to the wild type, despite its impaired cAMP responsiveness. The N199Q mutant, however, showed low pERK1/2 activity at both 5 and 15 minutes. Notably, while the N268Q and N293Q mutants showed pERK1/2 activity comparable to eFSHR-wt at 5 minutes, neither mutant exhibited any signaling at 15 minutes, despite their high cAMP responsiveness.
The abstract could be much more readable and clear. A large number of abbreviations and technical expressions can be discouraging for the reader, especially for those from the field of practical and applied sciences and clinical veterinary medicine. From the point of view of veterinary medicine, it is necessary to explain why these receptors are important and give specific examples of diseases or disorders. This will increase the visibility of this work beyond the framework of molecular biology. It is necessary to add one more sentence at the end and give you a more general and applied conclusion, that is, a sentence that will be the inductive conclusion of the described mechanism. This also applies to the conclusion in general, it is necessary to add a little more induction and deduction in order to understand the significance of these results in real life. I believe that the work is significant for the field of animal science and veterinary medicine, but it should be clear from the first reading. The introduction is well written. The aim of the research is well stated. It is not clear why an extremely large number of references were used in the introduction? Many of them are further used in the discussion. I think it is necessary to choose references adequately. MandM are written in too much tehnical detail and very much stifle the entire manuscript. Chapter 2.1 is redundant and should be deleted. It is necessary to describe the materials in accordance with methodological procedures where it is logical. Data analysis is correct. The results are adequately presented. The discussion is too extensive with a large number of detailed explanations and repetitions from the introduction. The conclusion is only of a technical nature, with no implications for the real significance of these results. It will not be clear to the average reader whether and which living tissues or cells were used. It is also not clearly visible whether there is permission from the ethics committee for this research. References are adequate.
Author Response
Reviewer 3
Comments and Suggestions for Authors
Thank you for the opportunity to review this manuscript.
The follicle-stimulating hormone receptor (FSHR) is a member of the G protein-coupled receptor (GPCR) superfamily, a vast and diverse group of cell surface proteins critical for signal transduction. GPCRs, including FSHR, are vital for mediating cellular responses to a wide array of agonists, thereby playing a central role in numerous physiological processes. They initiate receptor-mediated signaling cascades, notably the cyclic adenosine monophosphate (cAMP) pathway and the activation of phosphate extracellular signal-regulated kinases 1 and 2 (pERK1/2). Disruptions in these signaling pathways, often stemming from naturally occurring constitutively activating or inactivating mutations in FSHR, have been implicated in various diseases. In essence, the proper functioning of FSHR is essential for maintaining cellular homeostasis, and aberrations in its structure or signaling capabilities can have significant pathological consequences. Using site-directed mutagenesis, researchers created eFSHR mutants with mutations at each of the four N-linked glycosylation sites (N191Q, N199Q, N268Q, and N293Q). Wild-type (wt) eFSHR exhibited a dose-dependent increase in cAMP response, with a strong response observed at the EC50 and Rmax. In contrast, the N191Q and N199Q mutants showed a significant reduction in cAMP response. Compared to eFSHR-wt, the EC50 values for these mutants were approximately 0.46- and 0.44-fold, respectively, while the Rmax levels were 0.29- and 0.45-fold lower, respectively, even with high-ligand treatment. Conversely, the N268Q mutant displayed increased cAMP responsiveness, with EC50 and Rmax values increased by 1.23- and 1.46-fold, respectively, compared to eFSHR-wt. In eFSHR-wt cells, pERK1/2 activity was rapid, peaking within 5 minutes and remaining sustained until 15 minutes before declining sharply. The N191Q mutant exhibited a similar pERK1/2 activation pattern to the wild type, despite its impaired cAMP responsiveness. The N199Q mutant, however, showed low pERK1/2 activity at both 5 and 15 minutes. Notably, while the N268Q and N293Q mutants showed pERK1/2 activity comparable to eFSHR-wt at 5 minutes, neither mutant exhibited any signaling at 15 minutes, despite their high cAMP responsiveness.
The abstract could be much more readable and clearer. A large number of abbreviations and technical expressions can be discouraging for the reader, especially for those from the field of practical and applied sciences and clinical veterinary medicine. From the point of view of veterinary medicine, it is necessary to explain why these receptors are important and give specific examples of diseases or disorders. This will increase the visibility of this work beyond the framework of molecular biology. It is necessary to add one more sentence at the end and give you a more general and applied conclusion, that is, a sentence that will be the inductive conclusion of the described mechanism. This also applies to the conclusion in general, it is necessary to add a little more induction and deduction in order to understand the significance of these results in real life. I believe that the work is significant for the field of animal science and veterinary medicine, but it should be clear from the first reading. The introduction is well written. The aim of the research is well stated. It is not clear why an extremely large number of references were used in the introduction? Many of them are further used in the discussion. I think it is necessary to choose references adequately. MandM are written in too much technical detail and very much stifle the entire manuscript. Chapter 2.1 is redundant and should be deleted. It is necessary to describe the materials in accordance with methodological procedures where it is logical. Data analysis is correct. The results are adequately presented. The discussion is too extensive with a large number of detailed explanations and repetitions from the introduction. The conclusion is only of a technical nature, with no implications for the real significance of these results. It will not be clear to the average reader whether and which living tissues or cells were used. It is also not clearly visible whether there is permission from the ethics committee for this research. References are adequate.
- A large number of abbreviations
→ We rewrote it in order to minimize abbreviations in all manuscript including Title as much as possible.
→ Abbreviations were used only for commonly accepted terms. For example, eFSHR, LH/CGR, CG, FSH, TSH, GPCR, cAMP, ECâ‚…â‚€, Rmax, pERK1/2, PKA, Raf, MEK, and wt were presented in their abbreviated forms.
- it is necessary to explain why these receptors are important and give specific examples of diseases or disorders
→ We inserted “Glycosylation regulates G protein-coupled receptor mechanisms by influencing folding, ligand binding, signaling, trafficking, and internalization” in the Abstract part.
- It is necessary to add one more sentence at the end and give you a more general and applied conclusion
→ We inserted “Our results provide strong evidence for a new paradigm in which cAMP signaling is not activated, yet pERK1/2 cascade remains strongly induced” at the end of Abstract
- It is not clear why an extremely large number of references were used in the introduction? Many of them are further used in the discussion
→ We checked “introduction’ reference number” and moved to discussion section. And some references were removed.
- Chapter 2.1 is redundant and should be deleted
→ We removed the repeated materials based on the reviewer's comments and reduced the overall size.
- The discussion is too extensive with a large number of detailed explanations and repetitions from the introduction.
→ We rechecked the discussion section and deleted repetitions with introduction.
- The conclusion is only of a technical nature, with no implications for the real significance of these results.
→ We rewrote “Conclusion section”.
- It is also not clearly visible whether there is permission from the ethics committee for this research.
→ Our research does not include an ethics committee approval. However, This experiment adhered to the fundamental experimental guidelines.

Round 2
Reviewer 1 Report
Comments and Suggestions for Authors
The authors have incorporated most of the suggested changes.
Line 313: The N191Q and N199Q mutants showed a 63% and 40% decrease in pERK1/2 levels at 15 min, respectively. The author has not corrected this statement. Please correct as “The N191Q and N199Q mutants showed a decrease to 63% and 40% in pERK1/2 levels at 15 min, respectively.”
The authors said that they did not present analyses of cell surface expression levels or cell surface loss experiments. It’s fine; my concern is, did they ever check the protein expression of those clones after making it? How is it that they have not run even a single western blot to check the expression of clones? A simple western blot analysis will show that all the clones are equally expressed relative to the wt. When a functional analysis for any clone is performed, it must be done after checking the expression of those clones.
Author Response
Reviewer 1
The authors have incorporated most of the suggested changes.
Line 313: The N191Q and N199Q mutants showed a 63% and 40% decrease in pERK1/2 levels at 15 min, respectively. The author has not corrected this statement. Please correct as “The N191Q and N199Q mutants showed a decrease to 63% and 40% in pERK1/2 levels at 15 min, respectively.”
→ We rechecked it and revised the sentence based on the reviewer's comment.
The authors said that they did not present analyses of cell surface expression levels or cell surface loss experiments. It’s fine; my concern is, did they ever check the protein expression of those clones after making it? How is it that they have not run even a single western blot to check the expression of clones? A simple western blot analysis will show that all the clones are equally expressed relative to the wt. When a functional analysis for any clone is performed, it must be done after checking the expression of those clones.
→ Based on your comments, we performed the western blot analysis 2–3 times. however, we did not detect a specific band in the mutants, likely due to low expression. We were unable to perform additional experiments in time to submit this paper by early February. However, in another manuscript on equine LH/CGR currently under preparation, we present results from ELISA analysis of expression levels and a loss-of-receptor experiment. We performed the transfection using a transient method; therefore, we did not isolate any clones.

Reviewer 2 Report
Comments and Suggestions for Authors
The revised manuscript will be acceptable in its current form, based on the comments from authors.
Author Response
Reviewer 2
The revised manuscript will be acceptable in its current form, based on the comments from authors.
→Thanks very much for your fast reviewer

Reviewer 3 Report
Comments and Suggestions for Authors
Thank you.
Author Response
Reviewer 3
Comments and Suggestions for Authors
Thank you.
→Thanks very much for your fast reviewer

Round 3
Reviewer 3 Report
Comments and Suggestions for Authors
High percent of similarity in iThenticate. Please discuss with editors.